# Thermochromic Behavior of VO$_2$/Polymer Nanocomposites for Energy Saving Coatings

**Michalis Xygkis** [1,2], **Emmanouil Gagaoudakis** [1,2], **Leila Zouridi** [1,3], **Olga Markaki** [1,2], **Elias Aperathitis** [1], **Kyriaki Chrissopoulou** [1], **George Kiriakidis** [1,2] and **Vassilios Binas** [1,4,*]

1   Institute of Electronic Structure and Laser, Foundation for Research and Technology Hellas 100 N, Plastira str., Vassilika Vouton, 70013 Heraklion, Crete, Greece; ph4213@edu.physics.uoc.gr (M.X.); mgagas@iesl.forth.gr (E.G.); l.zouridi@iesl.forth.gr (L.Z.); ph4308@edu.physics.uoc.gr (O.M.); eaper@iesl.forth.gr (E.A.); kiki@iesl.forth.gr (K.C.); kiriakid@iesl.forth.gr (G.K.)
2   Department of Physics, University of Crete, 71003 Heraklion, Crete, Greece
3   Department of Material Science & Technology, University of Crete, 71003 Heraklion, Crete, Greece
4   Department of Physics, Crete Center for Quantum Complexity and Nanotechnology, University of Crete, 71003 Heraklion, Crete, Greece
*   Correspondence: binasbill@iesl.forth.gr; Tel.: +2810-391269

**Abstract:** Vanadium dioxide (VO$_2$) is a well-known thermochromic material that can potentially be used as a smart coating on glazing systems in order to regulate the internal temperature of buildings. Most growth techniques for VO$_2$ demand high temperatures (>250 °C), making it impossible to comply with flexible (polymeric) substrates. To overcome this problem, hydrothermally synthesized VO$_2$ particles may be dispersed in an appropriate matrix, leading to a thermochromic coating that can be applied on a substrate at a low temperature (<100 °C). In this work, we reported on the thermochromic properties of a VO$_2$/Poly-Vinyl-Pyrrolidone (PVP) nanocomposite. More specifically, a fixed amount of VO$_2$ particles was dispersed in different PVP quantities forming hybrids of various VO$_2$/PVP molar ratios which were deposited as films on fused silica glass substrates by utilizing the drop-casting method. The crystallite size was calculated and found to be 35 nm, almost independent of the PVP concentration. As far as the thermochromic characteristics are concerned, the molar ratio of the VO$_2$/PVP nanocomposite producing VO$_2$ films with the optimum thermochromic properties was 0.8. These films exhibited integral solar transmittance modulation (overall wavelengths) $\Delta Tr_{sol}$ = 0.35%–1.7%, infrared (IR) switching at 2000 nm $\Delta Tr_{IR}$ = 10%, visible transmittance at 550 nm $Tr_{Vis}$ = 38%, critical transition temperature $T_C$ = 66.8 °C, and width of transmittance hysteresis loop $\Delta T_C$ = 6.8 °C. Moreover, the critical transition temperature was observed to slightly shift depending on the VO$_2$/PVP molar ratio.

**Keywords:** hydrothermal synthesis; thermochromic VO$_2$; low-temperature VO$_2$/PVP nanocomposites; thermochromic coatings

## 1. Introduction

Versatile solutions to the expanding and rapid increase in worldwide energy demands as well as the restrictions concerning environmental pollution by utilizing renewable energy sources and energy-efficient materials have already attracted both scientific interest and commercial attention. The energy consumption in buildings alone is estimated to be approximately 40% of the world's total energy consumption, and it is anticipated it will increase steadily [1–3]. Central heating, ventilation, and air conditioning are the main energy consumers, being responsible for about 30% of annual carbon dioxide emissions [3]. Moreover, heat transmittance and insulation inefficiency of windows are responsible for 15%–22% of a building's energy loss [4–7].

In recent years, 'smart' windows have received widespread attention as one of the potential solutions to reducing energy consumption by air conditioning in modern architecture. 'Smart' windows are able to intelligently self-regulate the amount of transmitted heat, while keeping the visible transmission mainly unchanged. Vanadium dioxide ($VO_2$) is one of the most promising solid-sate materials for smart windows due to its unique optical properties related to its inherent and ultrafast reversible structural (phase-change) transition from a monoclinic $VO_2(M)$ to tetragonal rutile $VO_2(R)$ structure at a critical transition temperature of $T_C$ = 68 °C, for pure monocrystalline material [8–10].

Conventionally, 'smart' thermochromic windows are fabricated by vapor phase deposition techniques such as sputtering [11–14], chemical vapor deposition (CVD) [15–17], and pulsed laser deposition (PLD) [18]. However, all these techniques are restricted by the cost and scale of vacuum systems. Moreover, most of these growth techniques, although they are of high fidelity and can produce good quality thermochromic $VO_2$ films, demand high deposition temperatures (>400 °C) [11,14–18], with very few employing sputtering techniques between 250 and 300 °C [12,13,19–21], making it impossible to utilize flexible (polymeric) substrates.

Another approach, called the ex-situ approach [22], to the fabrication of $VO_2$ thermochromic films is to first synthesize the desired material as a powder and then to deposit the material as a film onto the desired surface. Thermochromic $VO_2$ in the form of powders have been synthesized by various methods, such as thermolysis [23,24], rapid thermal annealing [25], pyrolysis [26], and, the most utilized method, hydrothermal (solvothermal) synthesis [27–31]. The latter is the most promising due to the high crystallinity of the resulting products, the precise phase control, the versatility on the synthetic parameters, and the possibility of large-scale synthesis [32].

In order to transform the powder into film, various deposition methods have been applied, most notably sol-gel methods. Sol-gel is a proper inexpensive method for large-scale deposition, utilizing fast processing techniques such as dip and spin coating [33]. In particular, the sol-gel method is suitable for the deposition of $VO_2$, either directly on rigid or on flexible substrates (membrane) attached to glass surfaces. In the sol-gel method, a mixture of a precursor-supporting material along with the particles of the functional material are used (e.g., usually a polymer or monomer along with $VO_2$ particles) in order to synthesize the films [33–35]. Hence the polymer has to be adequately selected to fulfill several demands, including: (i) Stabilization of the $VO_2$ particles dispersion, (ii) prevention of agglomerations, and (iii) protection of $VO_2$ particles, e.g., from oxidation/reduction by air or water vapor, which is especially important during the long storage period. Among the various prospective polymeric host matrices, such as polyoxymethylene, polyvinyl alcohol, polyacrylic acid, etc., the one fulfilling the above criteria to a fairly satisfying level has been identified to be Poly-Vinyl-Pyrrolidone (PVP) [36]. Although the sol-gel process is a promising technique for achieving a low deposition temperature for thermochromic $VO_2$ film fabrication, various concerns such as thickness control and repeatability have not yet been met equivalently in all the different deposition methods used (spin-coating, drop-casting, dip-coating, and spraying), thus sustaining the interest of the research community and industry [37,38].

In the present work, thermochromic $VO_2$/polymer nanocomposite films were deposited uniformly using a casting method on glass substrates. The method selected was polymer-assisted deposition employing the drop-casting technique. The polymer utilized was PVP. A parametric study was carried out in order to examine the effect of the $VO_2$/PVP molar ratio on the thermochromic properties. The quantities of both the dispersant (distilled water) and the $VO_2$ particles were kept constant, while the quantity of PVP was varied, in order to change the molar ratio of $VO_2$/PVP. It is the first time that such a study has been performed, since until now the majority of research [39–44] has mainly focused on the thermochromic behavior of $VO_2$ powder/composite, without showing how the thermochromic properties are affected by the presence of a host material.

## 2. Materials and Methods

### 2.1. Synthesis of $VO_2$ Particles

The reagents used were vanadium pentoxide, $V_2O_5$ (+98% pure), as the vanadium source and oxalic acid dihydrate, $H_2C_2O_4 \cdot 2H_2O$ (≥99.0% pure), as the reducing agent. All reagents were purchased from Sigma Aldrich (Merck, Germany) and were used without further purification.

$VO_2$ particles were synthesized via a typical hydrothermal procedure using a Parr Teflon-lined stainless steel autoclave. In a typical procedure, 2 mmol of $V_2O_5$ powder and 8 mmol of $H_2C_2O_4 \cdot 2H_2O$ were dissolved in 2.22 mol of deionized water, resulting in a 32 vol% filling of the Teflon vessel. After stirring, the original dark yellow mixture turned into a dark green-blue solution. For the hydrothermal treatment, the precursor mixture was transferred to the acid digestion vessel and into a furnace for treatment at 220 °C for 12 h. The obtained blue-black solid product was isolated via centrifugation, after drying at 80 °C for 4 h. Finally, to acquire the desired crystalline phase of thermochromic $VO_2$ the solid product was annealed at 700 °C for 2 h under a constant nitrogen gas flow.

### 2.2. $VO_2$/Polymer Nanocomposite Coatings

The as-prepared $VO_2$ particles were dispersed ultrasonically in deionized water at a concentration of 10 mg/mL for 30 min and stirred for 1 h to produce a suspension. Then, an appropriate amount of PVP K-30 was added in different molar ratios of $VO_2$/PVP, from 0.2 to 1.2, by decreasing the amount of PVP while keeping constant the amount of $VO_2$ particles. The suspension was stirred for 2 h to enhance its homogeneity. Prior to the deposition, fused silica glass substrates were treated with ethanol and propanol inside a sonication bath for 5 min, respectively. Finally, the mixture was uniformly casted on the substrates and dried for 1 h at 70 °C.

### 2.3. Structural and Morphological Characterization/Thermochromic Properties

The structural characterization of $VO_2$ particles and nanocomposite films was performed by X-ray diffraction (XRD) using a Rigaku RINT-2000 diffractometer (Tokyo, Japan). The X-rays were produced by a 12 kW rotating anode generator with a Cu anode equipped with a secondary pyrolytic graphite monochromator. A Cu Kα radiation with wavelength λ = 0.154 nm was used. Measurements were performed with θ/2θ configuration, scanning from 20° to 80° with a step of 0.02°/min. The crystallite size of the $VO_2$ was calculated by using the Scherrer Equation;

$$d(\text{nm}) = \frac{0.9 \cdot \lambda(\text{nm})}{B \cdot \cos(\theta_B)} \qquad (1)$$

where λ = 0.154 nm, B the full width at half maximum (FWHM) at 2θ = 27.8° corresponding to the characteristic (011) direction of the $VO_2$ peak, and $\theta_B = \theta$.

The morphology of the $VO_2$ particles and nanocomposite films was determined by scanning electron microscopy (SEM, JEOL 7000, Tokyo, Japan) operating at 15 keV, while the microscopic nanostructures were observed by transmission electron microscopy (TEM) on a JEM-2100 instrument (JEOL, Tokyo, Japan) equipped with $LaB_6$ filament, operating at 200 kV.

The critical transition temperature of the $VO_2$ particles was determined by differential scanning calorimetry (DSC) using a PL-DSC system from Polymer Laboratories. Measurements were performed from 20 to 170 °C and back to 20 °C with a step of 10 °C/min, under a nitrogen flow of 20 cc/min. By identifying the critical transition temperature during heating ($T_1$) and cooling ($T_2$) procedures, the critical transition temperature ($T_C$) as well as the width of the hysteresis loop of the $VO_2$ powder were calculated as defined by the Equations below:

$$T_C = \frac{T_1 + T_2}{2} \qquad (2)$$

$$\Delta T_C = T_1 - T_2 \tag{3}$$

The thermochromic properties of the nanocomposite films were examined by recording the transmittance spectra in the temperature range of 25 and 90 °C. For this, a Perkin Elmer Lambda 950 UV/Vis/NIR spectrophotometer, operating at $\lambda = 250$–2500 nm with a homemade heating sample stage attachment and a thermocouple on contact with the surface of the film to measure the temperature, was used. The temperature was controlled by an ACUSHNET FP-900 temperature controller (Taipei, Taiwan) at a heating rate of 1.5 °C/min. IR switching is defined as the difference of transmittance value at $\lambda = 2000$ nm, between 25 and 90 °C.

$$\Delta Tr_{IR} (\%) = Tr_{IR}(25\,°C) - Tr_{IR}(90\,°C) \tag{4}$$

while integral luminous transmittance is defined as:

$$Tr_{lum}(\%) = \frac{\int_{350}^{750} B_{lum}(\lambda) * Tr(\lambda)d\lambda}{\int_{350}^{750} B_{lum}(\lambda)d\lambda} \tag{5}$$

where $Tr(\lambda)$ is the recorded transmittance spectrum at 25 or 90 °C and $B_{lum}(\lambda)$ is the standard luminous efficiency function for photopic vision [45].

Moreover, solar transmittance modulation $\Delta Tr_{sol}$ (%) is defined as the difference of integral solar transmission between 25 and 90 °C, obtained by the formula:

$$\Delta Tr_{sol}(\%) = \frac{\int_{250}^{2500} B_{sol}(\lambda) * Tr(T = 25\,°C, \lambda)d\lambda - \int_{250}^{2500} B_{sol}(\lambda) * Tr(T = 90\,°C, \lambda)d\lambda}{\int_{250}^{2500} B_{sol}(\lambda)d\lambda} \tag{6}$$

where $Tr(\lambda)$ is the recorded transmittance spectrum at 25 or 90 °C and $B_{sol}(\lambda)$ is the solar irradiance spectrum for an air mass (AM) of 1.5 (corresponding to a solar zenith angle of 48.2° [46].

Furthermore, in order to determine the critical transition temperature of $VO_2$/polymer nanocomposite films through optical measurements, the transmittance as a function of temperature (25–90 °C) by a step of 1.5 °C/min at $\lambda = 2000$ nm was recorded for both the heating and cooling procedure. The derivative of transmittance ($dTr/dT$) versus temperature was plotted and fitted by a Gaussian curve for both procedures. The minimum of the fitting curves determined the critical transition temperature $T_1$ and $T_2$ for the heating and cooling procedure, respectively. Thus, the critical transition temperature $T_C$ and the width of transmittance hysteresis loop were obtained by Equations (2) and (3), respectively.

## 3. Results and Discussion

### 3.1. Structural and Morphological Characterization of VO₂ Particles

Figure 1a shows the XRD pattern of $VO_2$ powder particles (for $2\theta = 20$–80°), from which the monoclinic phase of $VO_2$ (M) is confirmed (JCPDS card No. 43-1051). The crystallite size of the $VO_2$ powder was calculated from the XRD pattern using Scherer's Equation (Equation (1)) and was found to be around 35 nm. TEM imaging revealed the material to be composed of nanoflakes, with particles varying from less than 25 to 400 nm, as presented in Figure 1e,f. The thermochromic behavior of the $VO_2$ particles was confirmed by the DSC diagram of Figure 1b, from which the critical transition temperature and the hysteresis width were calculated and found to be $T_C = 66.6$ °C and $\Delta T_C = 12.1$ °C. The morphology of the $VO_2$ particles was investigated by SEM, presented in Figure 1c,d, in which nanoflake agglomerates with sizes up to about 500 nm can be observed.

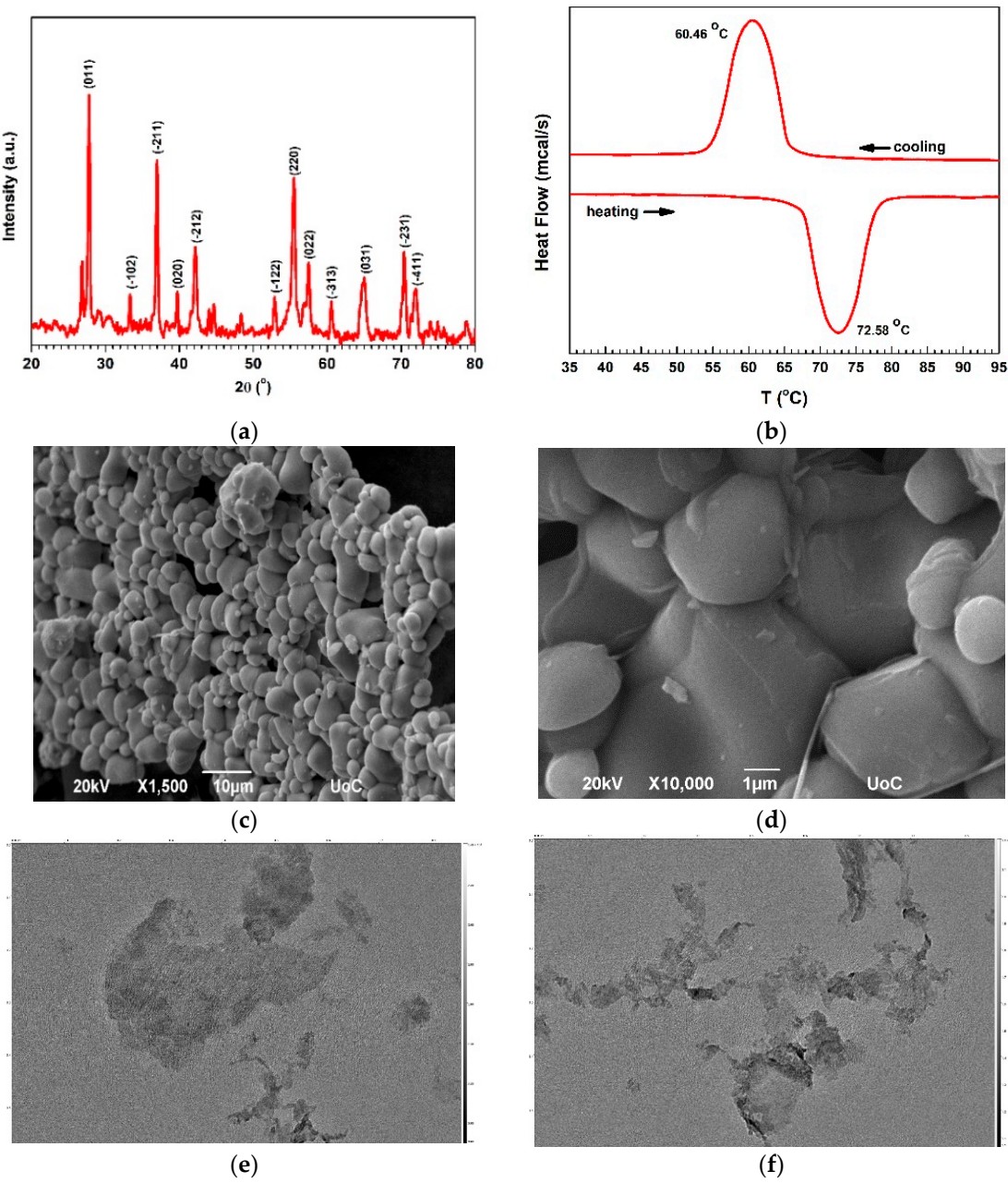

**Figure 1.** (**a**) XRD pattern; (**b**) differential scanning calorimetry (DSC) curve; (**c**,**d**) SEM images, and (**e**,**f**) TEM images of VO$_2$ powder particles.

*3.2. Structural and Morphological Characterization of VO$_2$/Polymer Nanocomposite Films*

The XRD pattern of VO$_2$/polymer nanocomposite films at the optimized ratio of VO$_2$/PVP = 0.8 deposited on fused silica glass substrate is shown in Figure 2a, along with a picture of the film in Figure 2b. The monoclinic phase of VO$_2$ was confirmed by the two characteristic peaks at 2θ = 27.8° and 37.0° corresponding to the VO$_2$ (011) and (200) crystallographic directions, respectively, according to JCPDS card No. 44-0252. The small peak at around 25° is attributed to the glass substrate. Thus, the polymer PVP does not affect the crystallinity of the VO$_2$. In addition, the crystallite size of VO$_2$ in the composite was calculated and found to be the same (35 nm) as that of VO$_2$ in powder form. However, the rest of the peaks corresponding to the VO$_2$ powder (Figure 1a) could not be detected in the composite, probably due to the presence of the host material (PVP).

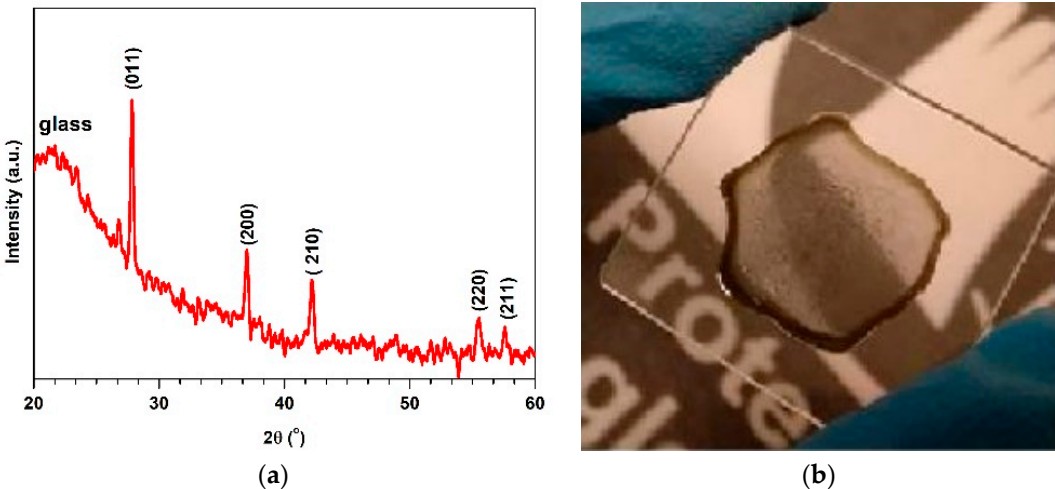

**Figure 2.** (**a**) XRD pattern and (**b**) image of $VO_2$/Poly-Vinyl-Pyrrolidone (PVP) nanocomposite film with ratio $VO_2$/PVP = 0.8 deposited on fused silica substrate.

In order to investigate the effect of the amount of PVP in the morphology of the surface of $VO_2$/polymer nanocomposite films, SEM at ×1500 magnification was performed. $VO_2$ particles, as shown in Figure 3, were found to have almost entirely been submerged, particularly at the optimized ratio of $VO_2$/PVP = 0.8, within the polymer matrices exhibiting a smooth composite finish surface without cracks. Moreover, it was found that the presence of polymer for all ratios from 0.2 to 1 prevented the agglomeration of particles with an average grain size of more than 500 nm.

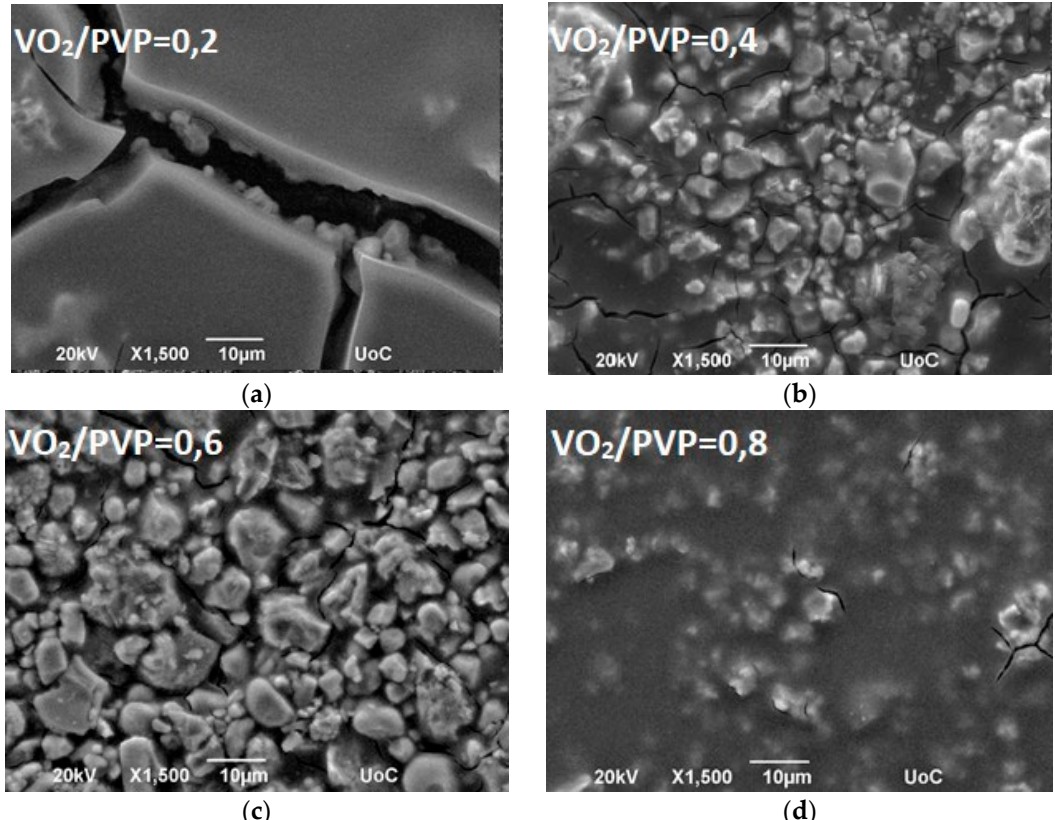

**Figure 3.** *Cont.*

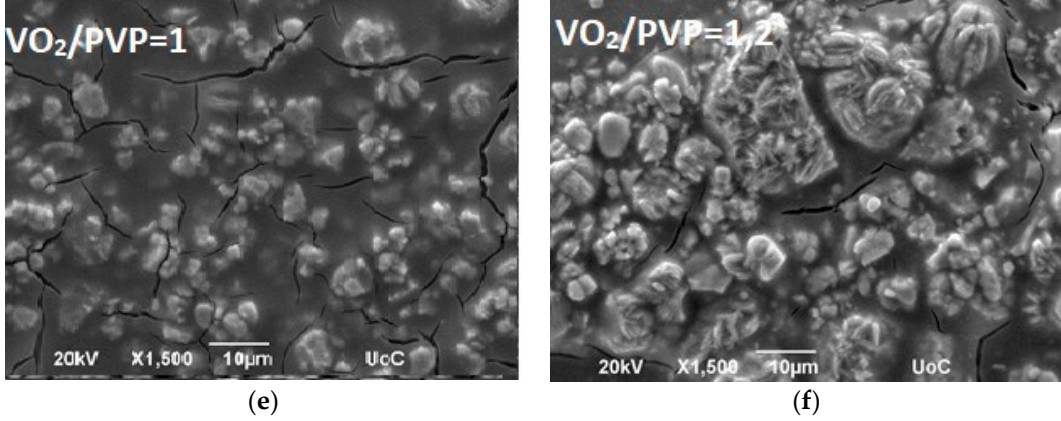

**Figure 3.** SEM images of VO$_2$/PVP nanocomposite films with a different molar ratio of VO$_2$: PVP from 0.2 to 1.2.

### 3.3. Effect of VO$_2$/PVP Molar Ratio on Thermochromic Properties

The transmittance spectra from 250 to 2500 nm of VO$_2$/polymer nanocomposite films with a different VO$_2$/PVP molar ratio at 25 and 90 °C are presented in Figure 4a–d. It can be seen that all films exhibit a thermochromic behavior. Utilizing these spectra along with Equations (4)–(6) IR switching at 2000 nm, $Tr_{lum}$ and $Tr_{sol}$, were calculated and are presented in Table 1.

**Table 1.** Optical characteristics of VO$_2$/polymer nanocomposite films.

| VO$_2$/PVP * | $Tr_{lum}$(%) | $\Delta Tr_{IR}$(%) | $\Delta Tr_{sol}$(%) |
|:---:|:---:|:---:|:---:|
| 0.2 | 24.5 | 7 | 1.72 |
| 0.4 | 27.2 | 5.7 | 0.36 |
| 0.6 | 26.9 | 7.8 | 1.00 |
| 0.8 | 37.1 | 10.2 | 1.22 |
| 1 | 36.8 | 7.8 | 1.35 |
| 1.2 | 41.2 | 4.8 | 0.35 |

\* molar ratio.

According to Table 1, it is clear that the luminous transmittance is decreasing as the concentration of PVP increases (i.e., the ratio of VO$_2$/ PVP is decreases) due to the anticipated increase of film thickness, varying between 3 and 5 µm, as a result of the casting method. Concerning the effect of PVP on IR switching, there is an optimal ratio of VO$_2$/PVP = 0.8, for which IR switching becomes maximum and equal to 10.2%, as presented in the graphical representation of data in Figure 5a. Furthermore, the calculated solar transmittance modulation varied from 0.35% to 1.7% without any evident indication on whether there is a correlation with the VO$_2$/PVP molar ratio, rather than a moderately noticeable non-linear increase of $\Delta Tr_{sol}$ with the increase of PVP in solution concentration (i.e., a decrease of the VO$_2$/PVP molar ratio). Although solar transmittance modulation is low enough when compared to thermochromic films prepared by other techniques at higher deposition temperatures [1,4,16], a systematic investigation on the dispersion of VO$_2$ powder in the solvent should lead to an improvement of thermochromic characteristics. Additionally, a different casting method such as dip coating or spin coating will be employed in order to increase the homogeneity of the films, resulting in an increase of luminous transmittance.

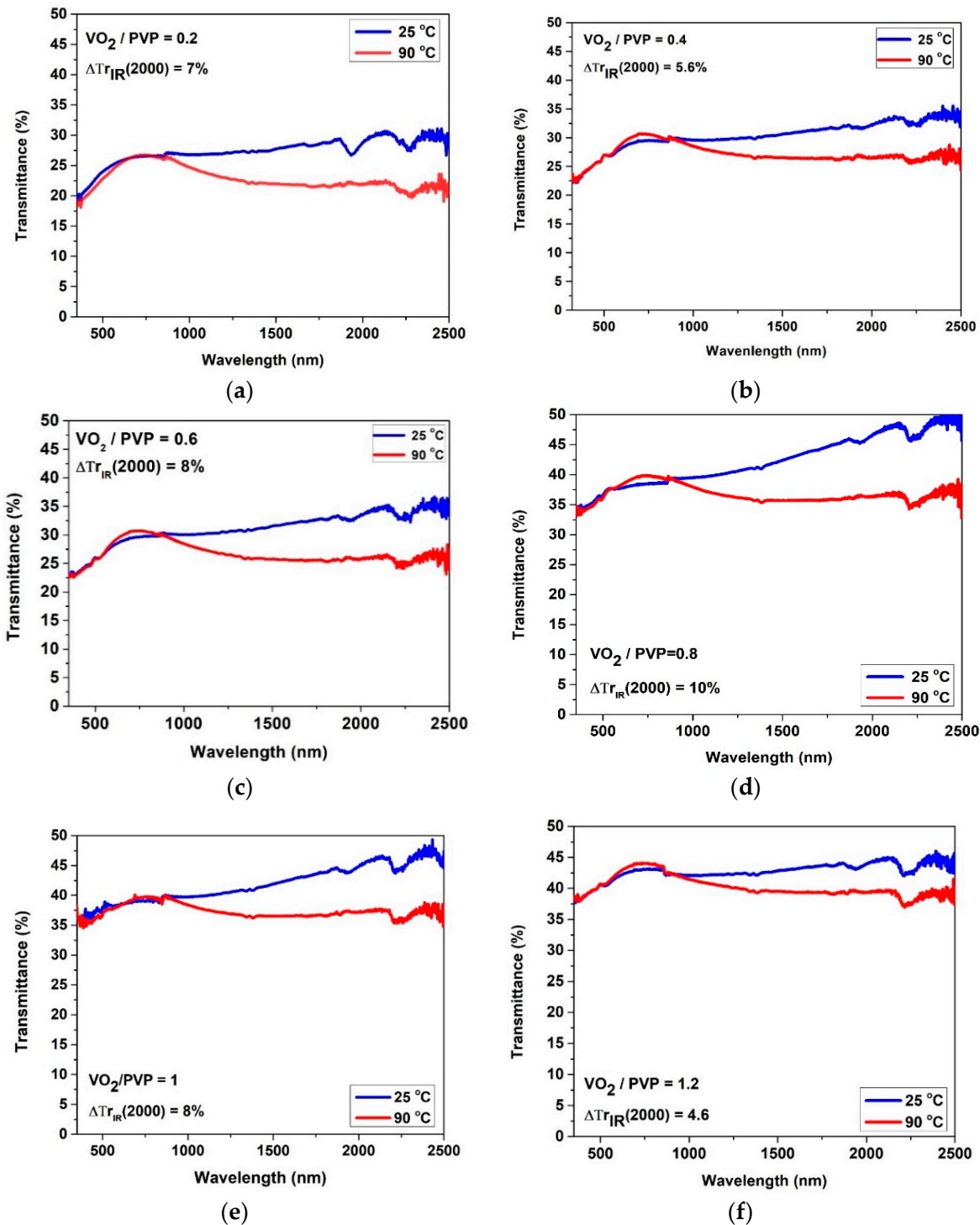

**Figure 4.** Transmittance spectra at 25 (blue line) and 90 °C (red line) of $VO_2$/PVP nanocomposite films prepared with different molar ratios of $VO_2$/PVP: (**a**) 0.2, (**b**) 0.4, (**c**) 0.6, (**d**) 0.8, (**e**) 1.0, and (**f**) 1.2.

The transmittance hysteresis loop of $VO_2$/PVP nanocomposites with different amounts of PVP, at $\lambda$ = 2000 nm is presented in Figure 6. From these heating–cooling loops, it is evident that the transition is a fully reversible process for all $VO_2$/PVP nanocomposite samples. By using Equations (2) and (3), the critical transition temperature and the hysteresis width, respectively, were calculated. The results are presented in Table 2, while in Figure 5b the variation of $T_C$ is graphically presented as a function of the $VO_2$/PVP molar ratio. It is observed that $T_C$ is increasing in regards to the molar ratio of $VO_2$/PVP from 62 °C ($VO_2$/PVP = 0.2) to 68 °C ($VO_2$/PVP = 1.2). This behavior can be attributed

to the increasing concentration of polymer in the films (from the 1.2 to the 0.2 $VO_2$: PVP molar ratio), since a higher polymer concentration seems to inhibit the agglomeration of $VO_2$ nanoparticles. Thus, the transition is enhanced by reducing the energy needed for $VO_2$ particles in the film to transition from the monoclinic to the full tetragonal rutile phase (and vice versa), leading to lower $T_C$ values. A similar phenomenon is observed for the width of transmittance hysteresis loop, since it is increased from 3.5 °C for $VO_2$/PVP = 0.2 to 8.1 °C for $VO_2$/PVP = 1.2, indicating that the increase of PVP facilitates the transition, thus lowering the hysteresis width.

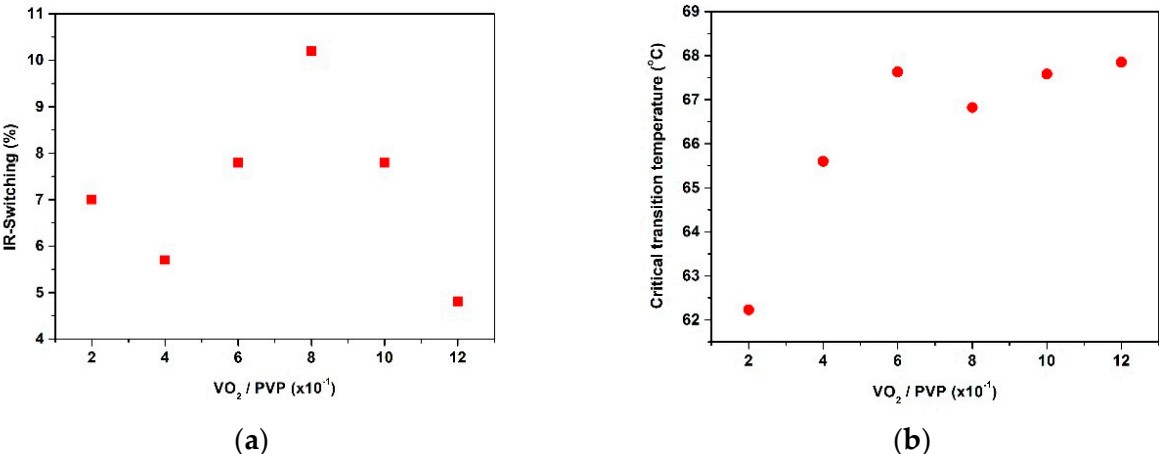

(**a**)   (**b**)

**Figure 5.** Effect of the $VO_2$/PVP molar ratio on thermochromic characteristics of $VO_2$/PVP nanocomposite films. (**a**) IR switching ($\Delta Tr_{IR}$), as calculated by Equation (4) at λ = 2000 nm and (**b**) critical transition temperature ($T_C$).

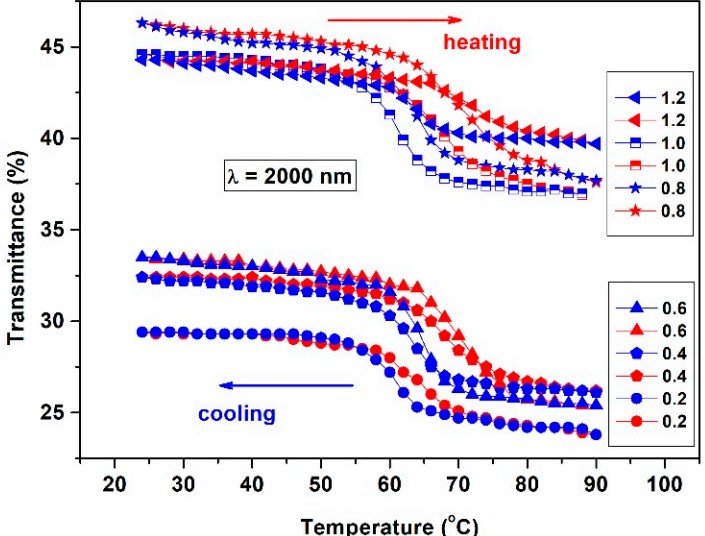

**Figure 6.** Transmittance hysteresis loop of $VO_2$/PVP nanocomposite films prepared with different amounts of PVP, recorded at λ = 2000 nm. Heating (red line), cooling (blue line).

**Table 2.** Critical transition temperature $T_C$ and width of transmittance hysteresis loop ($\Delta T_C$) of $VO_2/PVP$ nanocomposite films for different amounts of PVP.

| $VO_2/PVP$ * | $\Delta T_C$ (°C) | $T_C$ (°C) |
|---|---|---|
| 0.2 | 3.5 | 62.2 |
| 0.4 | 5.0 | 65.6 |
| 0.6 | 6.0 | 67.6 |
| 0.8 | 6.8 | 66.8 |
| 1.0 | 4.9 | 67.6 |
| 1.2 | 8.1 | 67.8 |

\* molar ratio.

## 4. Conclusions

In this work, hydrothermal synthesis was employed to formulate pure monoclinic $VO_2$ particles. Subsequently, $VO_2/PVP$ nanocomposite films were fabricated by polymer-assisted deposition on fused silica commercial glass at low treatment temperatures (<70 °C). Both thermochromic and optical properties were examined as a function of the $VO_2/PVP$ molar ratio varying from 0.2 to 1.2, by only changing the amount of PVP in the solution. The IR switching at 2000 nm was found to depend on the molar ratio of $VO_2/PVP$ and the optimal ratio was determined to be 0.8, for which $\Delta Tr_{IR} = 10.2\%$. The integrated luminous transmittance varied from 24.5% to 41.2% as the ratio of $VO_2/PVP$ increased from 0.2 to 1.2 (i.e., the amount of PVP was decreased), probably due to the increase on thickness of the produced films. Additionally, the solar transmittance modulation of the $VO_2/PVP$ nanocomposites non-linearly increased from 0.35% to 1.7% as the amount of PVP increased (i.e., a lower $VO_2/PVP$ molar ratio). Finally, it was observed that the critical transition temperature non-linearly increased from 62.2 to 67.8 °C as the ratio of $VO_2/PVP$ increased. This was attributed to the presence of the polymer which inhibits the agglomeration of the $VO_2$ particles, resulting in lower $T_C$ values for the transition.

**Author Contributions:** Conceptualization, E.G. and V.B.; Methodology, L.Z.; Software, M.X.; Validation, L.Z., E.G. and V.B.; Formal Analysis, M.X.; Investigation, O.M.; Data Curation, E.G.; Writing–Original Draft Preparation, M.X.; Writing–Review and Editing, G.K.; E.A. and K.C.; Visualization, E.G.; Supervision, V.B.; Project Administration, V.B.; Funding Acquisition, G.K.

**Funding:** This research was funded by a research grant from the Hellenic Ministry of Education with the acronym EXOTHERMO (09SYN-32-1185) and Innovation-EL, and the project "Electronics Beyond Silicon Era" (ELBYSIER) Erasmus+ KA2 programme.

**Acknowledgments:** The authors would like to deeply thank Lampros Papoutsakis and Dora Dragani for their time and effort, which proved crucial for the completion of this work.

**Conflicts of Interest:** The authors declare no conflict of interest. The funders had no role in the design of the study; in the collection, analyses, or interpretation of data; in the writing of the manuscript, and in the decision to publish the results".

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
