# Peer review of "Thermochromic Behavior of VO2/Polymer Nanocomposites for Energy Saving Coatings"

_coatings, doi:10.3390/coatings9030163_

Reviewer 1 Report

This paper details the use of poly(vinyl pyrrolidone) as an aid to dispersing monoclinic VO2 nanocrystals onto glass substrates.

The authors note that the low temperature of deposition is advantageous as the method can be used to deposit on polymeric substrates.

For me the most interesting find is the fact that the critical transition temperature shifted slightly depending on the VO2/PVP ratio in the film.

This paper is suitable for publication after addressing the following points:

-In the introduction, hydrothermal synthesis of VO2 should be mentioned as a synthesis technique, with examples in the following references:

[1]    M. J. Powell, P. Marchand, C. J. Denis, J. C. Bear, J. A. Darr, I. P. Parkin, Nanoscale 2015, 7, 18686–18693.
[2]    D. Malarde, I. D. Johnson, I. J. Godfrey, M. J. Powell, G. Cibin, R. Quesada-Cabrera, J. A. Darr, C. J. Carmalt, G. Sankar, I. P. Parkin, et al., J. Mater. Chem. C 2018, 6, 11731–11739.

Line 89- Should be "concentration"

-The use of the Scherrer equation to determine crystallite size is fine for an estimate, but I would thoroughly recommend transmission electron microscopy (TEM) be used to give the reader a more accurate picture of the particle size distribution.

Line 204- Should be subscript for "VO2"

-What is the relevance of figure 5? The caption needs more detailed and the axes should be labelled properly.

-What colour are the films? Having "yellow" windows is one of the chief reasons VO2 based thermochromic technology has not been taken up on a global scale.
- I would include photographs of as made films on glass substrates to give the reader an indication of the colour and visible transparency of the films.

Author Response

1.In the introduction, hydrothermal synthesis of VO2 should be mentioned as a synthesis technique, with examples in the following references:

[1]    M. J. Powell, P. Marchand, C. J. Denis, J. C. Bear, J. A. Darr, I. P. Parkin, Nanoscale 2015, 7, 18686–18693.
[2]    D. Malarde, I. D. Johnson, I. J. Godfrey, M. J. Powell, G. Cibin, R. Quesada-Cabrera, J. A. Darr, C. J. Carmalt, G. Sankar, I. P. Parkin, et al., J. Mater. Chem. C 2018, 6, 11731–11739.

Authors reply: DONE

2.Line 89- Should be "concentration"

Authors reply: DONE

The use of the Scherrer equation to determine crystallite size is fine for an estimate, but I would thoroughly recommend transmission electron microscopy (TEM) be used to give the reader a more accurate picture of the particle size distribution.

Authors reply: DONE. We add figure 1e and f.

4.Line 204- Should be subscript for "VO2"

Authors reply: DONE

5.What is the relevance of figure 5? The caption needs more detailed and the axes should be labelled properly.

Authors reply: DONE

6.What colour are the films? Having "yellow" windows is one of the chief reasons VO2 based thermochromic technology has not been taken up on a global scale.

Authors reply: DONE

7.I would include photographs of as made films on glass substrates to give the reader an indication of the colour and visible transparency of the films.

Authors reply: DONE. We add figure 2b.

Reviewer 2 Report

The manuscript deals with thermochromic VO2-based coatings prepared by sol-gel/hydrothermal technique. On the one hand, the manuscript deals with "hot" topic and employs a deposition technique which was not used so frequently for VO2. Thus, it may probably be eventually published.  On the other hand, the presentation should be significantly improved- please see the suggestions below.

(1) May be it is just a mistake/shift in numbering of references, but the references (I randomly checked only few of them) just do not contain the facts indicated by the text.
Example 1: Windows are responsible for 41 15–22% energy loss of a building [5]". ]". In fact, I see neither the interval 15-22 nor the word "window" in Ref. 5 ... please make clear which statement do you refer to.

Example 2: "techniques such as sputtering [7–9]". In fact, Refs. 7-9 are about VO2, but not really about sputtering of VO2. See the Introduction of e.g. doi.org/10.1016/j.solmat.2018.12.004 for a recent overview of truly relevant examples from various labs.

(2) The first part of the motivation (according to the Abstract) reads "Most of the growth techniques for VO2 demand high temperatures (>300°C) ... hydrothermally synthesized VO2 particles may be ... applied on a substrate at low temperature (<100 °C)". although="" i="" agree="" that="" qualitatively="" the="" motivation="" is="">300°C is slightly pesimistic. See e.g. doi.org/10.1016/j.tsf.2016.07.051 or doi.org/10.1088/1361-6463/aa8356 (exactly 300 °C in both cases), let alone even lower temperatures achieved using crystalline substrate (e.g. 250 °C doi.org/10.1016/j.apsusc.2016.10.084) or crystalline interlayer.

(3) The second part of the motivation (according to the Introduction) reads "Conventionally, smart thermochromic windows are fabricated by vapor phase deposition techniques such as sputtering ... CVD ... PLD ... However, all these techniques are restricted by the cost and scale of vacuum systems". I do not fully agree about the scale (some of the aforementioned techniques are used for window glasses, roll-to-roll deposition, etc.), but this is not the main point. The main point is that while I agree about the lower costs of sol-gel, it is necessary to make as clear as possible that the lower costs are achieved at preserved properties. The combination of luminous transmittance <=41.2% and solar transmittance modulation <=1.7% possibly constitutes a good first author's try using sol-gel, but does not seem to be competitive with the coatings (yes, prepared at higher temperature and costs) in the literature. I suggest to make the comparison with the literature, and discuss if and how the aforementioed numbers may be improved.

(4) The qualitative statements such as "due to the expected increasing of thickness" are OK, but I suggest to quantify the film thickness.

(5) Part of the worldwide efforts in this field are related to multilayer VO2-based coatings. This requires very precise thickness of the antireflection layers (+- few %), which may or may not be achievable using the author's deposition technique. Please discuss.

(6) The integral solar transmittance modulation (over all wavelengths) is more important that the IR switching (at a single wavelength), i.e. it would make sense to prefer the former in the abstract. Note that e.g. the aforementioned paper doi.org/10.1016/j.solmat.2018.12.004 shows that these two quantities are not even monotonically dependent for some coatings.

Author Response

1. May be it is just a mistake/shift in numbering of references, but the references (I randomly checked only few of them) just do not contain the facts indicated by the text.
Example 1: Windows are responsible for 41 15–22% energy loss of a building [5]". ]". In fact, I see neither the interval 15-22 nor the word "window" in Ref. 5 ... please make clear which statement do you refer to.

Example 2: "techniques such as sputtering [7–9]". In fact, Refs. 7-9 are about VO2, but not really about sputtering of VO2. See the Introduction of e.g. doi.org/10.1016/j.solmat.2018.12.004 for a recent overview of truly relevant examples from various labs.

Authors reply: DONE

2.The first part of the motivation (according to the Abstract) reads "Most of the growth techniques for VO2 demand high temperatures (>300°C) ... hydrothermally synthesized VO2 particles may be ... applied on a substrate at low temperature (<100 250="" 300="" .="" although="" i="" agree="" that="" qualitatively="" the="" motivation="" is="" c="" slightly="" pesimistic.="" see="" e.g.="" j.tsf.2016.07.051="" or="" aa8356="" exactly="" in="" both="" let="" alone="" even="" lower="" temperatures="" achieved="" using="" crystalline="" substrate="" span="" style="color: rgb(34, 34, 34)" .--="">

Authors reply: DONE

3. The second part of the motivation (according to the Introduction) reads "Conventionally, smart thermochromic windows are fabricated by vapor phase deposition techniques such as sputtering ... CVD ... PLD ... However, all these techniques are restricted by the cost and scale of vacuum systems". I do not fully agree about the scale (some of the aforementioned techniques are used for window glasses, roll-to-roll deposition, etc.), but this is not the main point. The main point is that while I agree about the lower costs of sol-gel, it is necessary to make as clear as possible that the lower costs are achieved at preserved properties. The combination of luminous transmittance <=41.2% and solar transmittance modulation <=1.7% possibly constitutes a good first author's try using sol-gel, but does not seem to be competitive with the coatings (yes, prepared at higher temperature and costs) in the literature. I suggest to make the comparison with the literature, and discuss if and how the aforementioed numbers may be improved.

Authors reply: DONE

4. The qualitative statements such as "due to the expected increasing of thickness" are OK, but I suggest to quantify the film thickness.

Authors reply: DONE

5.Part of the worldwide efforts in this field are related to multilayer VO2-based coatings. This requires very precise thickness of the antireflection layers (+- few %), which may or may not be achievable using the author's deposition technique. Please discuss.

Authors reply: DONE

6. The integral solar transmittance modulation (over all wavelengths) is more important that the IR switching (at a single wavelength), i.e. it would make sense to prefer the former in the abstract. Note that e.g. the aforementioned paper doi.org/10.1016/j.solmat.2018.12.004 shows that these two quantities are not even monotonically dependent for some coatings.

Authors reply: DONE

Reviewer 3 Report

In the manuscript, the authors prepared a VO2/PVP nanocomposite for thermal management applications. VO2 particles were previously synthesised in solution and then mixed with a commercially available PVP. The the composite was obtained via solution casting over a glass substrate and deeply characterised in terms of experiments to confirm and quantitatively evaluated the thermochromic behaviour. The manuscript contains interesting data and could be accepted in this journal after revision, according to the comments below:

1) I suggest the authors to revise the term "nano composite" since micro-sized particles of VO2 are present within the PVP matrix;

2) please revise the name of PVP with poly(vinyl pyrrolidone);

3) the supporting info file apparently results as the same of the main text;

4) the authors should provide comments of the the effect of the amount of PVP on the particles morphology. It results that chanced in particles size and distribution occurred with polymer content. Moreover, please report the thickness of the derive composite films and how it could affect the thermochromic behaviour;

5) I suggest to introduce better the chromogenic behaviour of VO2. It is reported that a phase change occurs at a certain temperature and it strongly affects the reflectance of the material. Based on that I suggest to add to the transmittance spectra the reflectance ones to better understand the chromogenic behaviour.

Author Response

1. I suggest the authors to revise the term "nano composite" since micro-sized particles of VO2 are present within the PVP matrix;

Authors reply: We put figure 1e and 1f with TEM image. As you can see its nano composite!

2. please revise the name of PVP with poly(vinyl pyrrolidone);

Authors reply: DONE

3. the supporting info file apparently results as the same of the main text;

Authors reply: we haven’t supporting info

4. the authors should provide comments of the the effect of the amount of PVP on the particles morphology. It results that chanced in particles size and distribution occurred with polymer content. Moreover, please report the thickness of the derive composite films and how it could affect the thermochromic behaviour;

Authors reply: DONE

5. I suggest to introduce better the chromogenic behaviour of VO2. It is reported that a phase change occurs at a certain temperature and it strongly affects the reflectance of the material. Based on that I suggest to add to the transmittance spectra the reflectance ones to better understand the chromogenic behaviour.

Authors reply: DONE

Round  2

Reviewer 2 Report

Although the response letter is not worth a lot (only done ... done ... done), the manuscript itself proves that the authors made sufficient efforts to address my comments. The manuscript can be published.

Reviewer 3 Report

In the revised version, the authors properly addressed most of the queries raised. Nevertheless, reflectance spectra still missing.